# Chemical Bonding in the C$_2$ Molecule

Alexander F. Sax 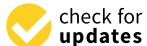

Department of Chemistry, University of Graz, Heinrichstrasse 28, 8010 Graz, Austria; alexander.sax@uni-graz.at

**Abstract:** Bonding in the C$_2$ molecule is investigated with CAS(8,8) wave functions using canonical MOs. In a subsequent step, orthogonal atomic orbitals are constructed by localizing the CASSCF MOs on the two carbon atoms with an orthogonal transformation. This orbital transformation causes an orthogonal transformation of the configuration state functions (CSF) spanning the function space of the singlet ground state of C$_2$. Instead of CSFs built from canonical MOs, one obtains CSFs of orthogonal deformed atomic orbitals (AO). This approach resembles the orthogonal valence bond (OVB) methods' CSFs, which are very different from the conventional VB, based on non-orthogonal AOs. To become used to the different argumentation, the bonding situations in ethane (single bond), ethene (double bond), and the nitrogen molecule (triple bond) are also studied. The complex bonding situation in C$_2$ is caused by the possibility to excite an electron with a spin flip from the doubly occupied 2$s$ AO into the 2$p$ subshell, and the resulting high-spin $^5S_u$ state of the carbon atom allows for a better reduction of the Pauli repulsion. However, the electron structure around the equilibrium distance does not allow one to say that C$_2$ in its ground state has a double, or triple, or even a quadruple bond.

**Keywords:** dicarbon; chemical bonding; spin reorganization; orthogonal valence bond

## 1. Introduction

Covalent bonding is a central concept in chemistry but its semantic is not unique. In physical parlance, bonding means the energetic stabilization of an unspecified size of a system composed of interacting subsystems by any kind of interactions [1] or any kind of attraction [2]; the result of bonding is a bonded system and often it is said that there is a bond in the stabilized system [2]. Depending on the amount of energy released during bonding, i.e., the bond energy, one can distinguish between weak (secondary) and strong (primary) bonding. In chemistry, chemical bonding is the thermodynamic stabilization of a molecular system at ambient conditions composed of atoms, free radicals, ions or molecules. Frequently, Coulomb interactions between negatively charged particles such as electrons or atomic anions and positively charged nuclei or atomic cations are said to be responsible for the stabilization of ionic solids, but even in such systems the Coulomb interaction is not solely responsible for it. Mainly responsible for the repulsion of many-electron ions is the fermionic character of the electrons, which prohibits identical electrons from coming too close. However, the bonding of non-charged subsystems needs a different description of system stabilization, called covalent bonding. The high reactivity of radicals with odd numbers of electrons and the observation that most stable molecules have an even number of electrons led Lewis to the assumption that linking radicals with one unpaired electron each yields a stabilized molecular system with an even number of electrons; this is the rule of two [3]. According to this view, the stabilization is caused by the formation of an electron pair that is shared by the atoms where the unpaired electrons are located. In the bonded system, these two atoms form a group with a characteristic short distance between them. This atom group with short distance is evidence for the formation of a covalent bond between the atoms. If more than one unpaired electron is located at each of the interacting atoms, multiple bonds can be formed. The number of bonds each atom can form is its

valence. However, this captivating Lewis model gives no convincing physical explanation for what causes the energetic stabilization.

The purely electrostatic model was first proposed by Slater [4] and is still most often used—not only in introductory textbooks—to explain covalent bonding. The line of reasoning is as follows: Electrons between the bonded atoms are attracted by both nuclei; if the electron density in the midbond region increases due to electron sharing, the (negative) potential energy and therefore also the total energy is lowered. However, this model does not agree with Earnshaw's theorem [5], which says that electrostatic interactions alone can never hold a system of charged particles in a stable, stationary state; the charges must be moving, and, therefore, the kinetic energy must play a central role in the stabilization of the system. According to Hellmann [6], the increase in the region of space where the shared electrons can be found causes a decrease in the kinetic energy and thus in the total energy, and this is the main reason for the stabilization of the system. The role of the kinetic energy was neglected for decades in both the physical and the chemical community. In 1962, Ruedenberg [7] demonstrated in a seminal paper that the kinetic energy is responsible for energetic stabilization, but it took another 40 years until Ruedenberg and coworkers [8–12] in a series of high level calculations could again substantiate the claim very convincingly. In addition, at the same time, they could also demonstrate that the energetic stabilization during covalent bonding is indeed a 1-electron effect, not a 2-electron effect.

The first molecule treated with quantum theory (Heitler and London, 1927) [13] was the hydrogen molecule: the archetypal molecule representing a covalent bond caused by an electron pair, as proposed by Lewis. The bonding electron pair was represented by the Heitler–London wave function $\Phi_{HL}$, which is the product of a two electron spatial wave function, also called a geminal, and the singlet spin function $\alpha(1)\beta(2) - \beta(1)\alpha(2)$. All wave functions that are eigenfunctions of the square of the spin operator are called configuration state functions (CSF); therefore, because of its product form, $\Phi_{HL}$ is a CSF. The spatial part of $\Phi_{HL}$ is a linear combination of products of atomic orbitals (AO), $1s_A(1)1s_B(2) + 1s_B(1)1s_A(2)$, each of the two hydrogen atoms A and B contributes one $1s$ AO (normalization factors are omitted). Because the Heitler–London CSF $\Phi_{HL}$ described the qualitatively correct bonding of two univalent hydrogen atoms by an electron pair, it was called the covalent wave function; the valence bond (VB) method uses wave functions that are generalizations of $\Phi_{HL}$. Around the same time, Hund and Mulliken supposed the existence of molecular orbitals (MO) in molecules similar to AOs in many-electron atoms [14–22]. For diatomic molecules, correlation diagrams correlating the orbital energies of the molecule to the orbital energies of the separated atoms and the united atoms allowed one to guess the energetic ordering of the MOs and to classify them as bonding and antibonding. Starting from the dissociated molecules, Lennard-Jones was the first to introduce positive and negative linear combinations of AOs to approximate MOs and to make the first quantitative calculation using MOs in the LCAO approximation [23]. The acronym LCAO (linear combination of atomic orbitals) was coined by Mulliken in 1932 [24]; the acronym is still used, although, in actual quantum chemical calculations not orbitals of free atoms but AO-like basis functions are used. The first quantitative SCF (self consistent field) calculation of the $H_2$ molecule with MOs was conducted by Coulson [25] using MOs in elliptic coordinates. However, it took another 20 years until SCF calculations with the LCAO approximation could be made [26,27]. The MOs were calculated as eigenfunctions of the Hermitian Hamiltonian and, thus, were known to be orthogonal to each other, which was not only a great computational advantage over the VB method but also had a great conceptual importance, e.g., an electron occupying a bonding MO can never occupy also an antibonding MO. Using a single Slater determinant $|\sigma\overline{\sigma}|$, which is also a CSF, with the doubly occupied bonding $\sigma$ MO, the bond energy of $H_2$ calculated by Goodisman [28] was about 0.5 eV lower than the bond energy calculated with $\Phi_{HL}$. On the other hand, it was also known that the single CSF $|\sigma\overline{\sigma}|$ gives reasonable results only for molecular structures close to the equilibrium geometry [29], but not when bonds are highly stretched; in contrast

to the VB method, the dissociated $H_2$ system has an energy that is much too high. Thus, both wave functions have deficiencies that must be corrected.

To improve the description of the ground state of $H_2$, Weinbaum [30] suggested to approximate the ground state wave function by a linear combination $\Psi_{W1} \propto \Phi_{HL} + \mu\Phi_{ion}$ of $\Phi_{HL}$ and the ionic CSF $\Phi_{ion} \propto 1s_A(1)1s_A(2) + 1s_B(1)1s_B(2)$ and demonstrated that optimization of the variation parameter $\mu$ improves the bond energy considerably. A second linear combination $\Psi_{W2}$, which can be made with $\Phi_{HL}$ and $\Phi_{ion}$, describes an excited state of $H_2$. In the dissociation limit, $\Phi_{HL}$ describes two neutral atoms, and thus is called a neutral CSF, whereas $\Phi_{ion}$ describes an anion-cation pair, and is accordingly termed an ionic CSF. However, when the interatomic distance is zero and atom A and atom B coalesce, not only the two different AOs but also the CSFs $\Phi_{HL}$ and $\Phi_{ion}$ become identical. This means, with decreasing interatomic distance, $\Phi_{HL}$ loses its neutral character and acquires ionic character; for $\Phi_{ion}$, it is the other way around.

If the LCAO approximation is used for the bonding MO $\sigma \propto 1s_A + 1s_B$, and if the CSF $|\sigma\bar{\sigma}|$ is expanded, one obtains a linear combination $|\sigma\bar{\sigma}| \propto \Phi_{ion} + \Phi_{HL}$ with equal coefficients of the linear combination for all bond lengths. It is this equal contribution of Heitler–London and ionic VB CSFs to the MO CSF $|\sigma\bar{\sigma}|$ that is the reason for the inability of the SCF wave function to describe the dissociation. The expansion of $|\sigma^*\bar{\sigma^*}|$ with the doubly occupied antibonding MO $\sigma^* \propto 1s_A - 1s_B$ again provides a linear combination of $\Phi_{HL}$ and $\Phi_{ion}$ with coefficients of equal modulus but now with a different relative phase, $|\sigma^*\bar{\sigma^*}| \propto \Phi_{ion} - \Phi_{cov}$. With a linear combination of these two MO CSFs, $\Psi_{MO1} \propto |\sigma\bar{\sigma}| - \lambda|\sigma^*\bar{\sigma^*}|$, $\lambda > 0$, the ionic contribution to the ground state wave function can be reduced and will disappear for long intermolecular distances; similarly, in a second linear combination $\Psi_{MO2}$, the neutral contributions to the excited state are reduced. Linear combinations of Slater determinants or CSFs are called CI (configuration interaction) wave functions; the variationally optimized CI wave functions $\Psi_{MO1}$ and $\Psi_{MO2}$ are equivalent with the optimized Weinbaum functions.

This shows that CI wave functions, both with VB and MO CSFs, provide qualitative correct and quantitative satisfying descriptions of the $H_2$ molecule and the dissociation reaction. The reason for the failure of the Slater determinant $|\sigma\bar{\sigma}|$ to describe correctly the dissociation is that two electrons occupying the bonding MO can never completely separate. In the ground state, each electron should locate during the dissociation at different atoms, and the corresponding electron distribution is correctly described by $\Psi_{HL}$, but not by $|\sigma\bar{\sigma}|$ because the form of the $\sigma$ MO forces the electrons to stay too close together. The antibonding MO $\sigma^*$, with the node between the atoms, describes an electron distribution where the electrons can never meet in the mid-bond region. The spatial correlation of the electrons in a midbond region is called a left–right correlation. Electrons that tend to stay on different sides of a plane in a molecule demonstrate angular correlation. To describe these two correlation types, only AOs of the valence shell are needed; a third correlation type, the in–out correlation, needs AOs with an additional radial nodal surface. Left–right and angular correlation contribute essentially to what is called non-dynamic correlation.

The most important reason for electron correlation is not charge redistribution caused by Coulomb interaction but the fermionic character of the electrons. The Pauli exclusion principle (PEP) says that identical electrons, which are electrons that agree also in the spin projection, avoid coming spatially close. Thus, if the total spin of the electrons in an atom changes from a low-spin state to a high-spin state, the electrons must locate in different spatial regions. Moreover, this spatial correlation of identical electrons is much more effective than the correlation due to the Coulomb repulsion. Non-dynamic correlation covers both spin redistribution and charge redistribution.

When two or more covalent bonds must be described, technical and conceptual problems emerge. Canonical MOs obtained as eigenfunctions of Fock operators are in general delocalized over the whole molecule; bonding between two neighboring atoms, which requires localized MOs, is hidden when canonical MOs are used. However, as elements of a vector space, orbitals can be linearly combined, and, therefore, local information can

be revealed by using proper transformations in the orbital space. The same is true for information hidden in CSFs made with canonical MOs. The methodological and technical improvements of quantum chemical methods in the last 60 years allowed for high-precision data such as bond energies or force constants, but the interpretation of bonding still uses concepts from the early times of theoretical chemistry. For example, the concept of bond order in MO theory is the half difference of the number of valence electrons occupying bonding MOs and the number of electrons occupying antibonding MOs; if the molecular state considered is represented by a wave function that consists of a single Slater determinant, the MOs can be occupied by two, one or zero electrons, and the bond order is an integer or half-integer. Although the description of chemical bonding is intrinsically multiconfigurational, for certain molecular geometries a single Slater determinant may dominate the multiconfigurational wave function, and only then the use of bond order to describe the bonding situation is justified. Also problematic is the identification of the Lewis electron pair with a covalent bond or the concept of a covalent bond with a Heitler–London geminal, or the identification of everything. The discussion about bonding in the $C_2$ molecule is paradigmatic for this semantic hodgepodge. In 2011, Shaik and Hiberty [31] published a VB study on $C_2$; in 2013, a quadruple bond was claimed in this molecule [32]. This was the starting point for a vivid discussion between critics of this claim [33–36] and its defenders [32,37–39]. The controversial issues were: (a) What contributions to the wave function must be counted to end up with four bonds; (b) how can one decide on whether or not an alleged bond is indeed a bond? The valence electron configuration in $C_2$ at the equilibrium geometry is assumed to be $2\sigma_g^2 2\sigma_u^2 \pi^4$ with energy-ordered MOs. The $\sigma$ MOs are linear combinations of sp hybrid AOs with the large lobe pointing towards to other C atom; because bonding and antibonding MOs are doubly occupied, this electron configuration describes a double bond made with two $\pi$ MOs. With the second set of sp hybrides with the lobes pointing away from the other C atom, another set of $\sigma$ MOs can be formed, and the electron configuration $2\sigma_g^2 \pi^4 3\sigma_g^2$ can be regarded as describing two $\sigma$ and two $\pi$ bonds; thus, it is a quadruple bond. Sherrill and Piecuch [40] demonstrated that CAS(8,8) wave functions cover all non-dynamic correlation effects in the $C_2$ molecule, which are essential for describing covalent bonding, and Frenking and Hermann [33] demonstrated that the CSFs corresponding to the first and the second electron configurations have weights of about 70% and 14%, respectively, in a CAS(8,8) wave function. Similarly, Xu and Dunning [41] using GVB (generalized VB) found that the perfect-pairing CSF, representing the four bonds by four singlet geminals, is not dominant in the wave function, and they simply stated: . . . *$C_2$ does not have a set of traditional covalent $\sigma$ and $\pi$ bonds such as $N_2$.* As one possible reason, it was assumed that the bonding properties of the $3\sigma_g$ MO are less pronounced than those of the $2\sigma_g$ MO, but this was not considered to corrupt the idea of a quadruple bond. After all, . . . *everyone in the discussion necessarily agrees that the $X\,^1\Sigma_g^+$ ground state of $C_2$ possesses four electron pairs* [37], but there was no agreement about the strengths of the bonds, even when one can speak of a bond. This is a fundamental semantic question. If one remembers that energy is the most important basis for scales in science, the question raised by Ramos-Cordoba et al., . . . *what measures do we accept to define what a "chemical bond" is?* Ref. [42], is indeed fundamental, but . . . *if there is electron pairing and if this brings about energy lowering more than a hydrogen bond, then we have a bond* [43,44] together with the variant *A bond is a bond is a bond* [32] of Gertrud Stein's famous phrase, is not a convincing answer. What puzzles me is the minor role that the fermionic character of electrons plays in this dispute. Pauli repulsion between electron pairs in bonds is discussed by Xu and Dunning [41]; Frenking mentions the Pauli principle between electrons with the identical spin projection [33]; it is discussed which state of the carbon atom is better suited as a reference state, the $^3P$ ground state of the carbon atom or the $^5S$ high-spin [32,33,35,45]. The discussion suggests that the authors expect that only either of the two atomic states can be reference states along the whole reaction coordinate. However, this would be a very shortsighted position, excluding the possibility of spin flips from low-spin to high-spin states in order to reduce any kind of repulsion between the electrons. If spin flipping does

occur one has to ask: what triggers it, is it geometry dependent, and does it occur in every state or is it symmetry dependent? These questions are in the center of this OVB analysis. There are indications for a preference of local high spin arrangements in the ground state around the local minimum, but the OVB—or any other—analysis of the electronic states of $C_2$ cannot provide a definite answer, because the carbon atoms in the molecule are entangled and thus are in mixed states. An expectation value of the local spin state needs the calculation of reduced density matrices for the subsystems considered, in this case of the carbon atoms.

## 2. Basics of OVB

The Heitler–London type geminal for $H_2$ $\Phi_{HL} = (1s_A(1)1s_B(2) + 1s_B(1)1s_A(2))/\sqrt{2(1+S^2)}$ with two hydrogen $1s$ AOs, which are in general not orthogonal to each other, $S = \langle 1s_A|1s_B \rangle \neq 0$, describes for only large interatomic distances a neutral electron distribution, i.e., two hydrogen atoms in their respective ground states. Moreover, for only the dissociated molecule, $\Phi_{HL}$ is orthogonal to the ionic geminal $\Phi_{ion} = (1s_A(1)1s_A(2) + 1s_B(1)1s_B(2))/\sqrt{2(1+S^2)}$, as can be observed from $\langle \Phi_{HL}|\Phi_{ion} \rangle = 2S/(1+S^2)$. When the interatomic distance goes to zero, $S$ goes to one and the overlap of the two geminals is one. That means $\Phi_{HL}$ and $\Phi_{ion}$ become linearly dependent; this poses problems for the interpretation of the VB CSFs: the characteristics of the geminals for the dissociated system are as different as can be, neutral vs. ionic, but at finite distances, especially around the equilibrium distance, $\Phi_{HL}$ has already acquired a substantial ionic character. This goes in parallel with the increase in the overlap integral, which is a measure of the superposition of the atomic states and thus of covalent bonding. That $\Phi_{HL}$ describes so well the ground state of $H_2$ [46] is the result of the increase in ionic character, and this makes $\Phi_{HL}$ indeed a covalent VB CSF. However, the change in the characteristics is hidden by the mathematical form of the geminal, which is the same for all interatomic distances. However, this constancy of the mathematical form of the geminal is often wrongly interpreted: *In the valence bond (VB) view. . . , the electrons are viewed to interact so strongly that there is negligible probability of finding two electrons in the same orbital. The wave function is thus considered to be dominated by purely covalent contributions in which each electron is spin-paired to another electron* [47]. Equating the mathematical form of a geminal with a certain physical interpretation is simply wrong.

If orthogonal AOs $1s_A$ and $1s_B$ are used, the CSFs $\Phi_{HL}^o$ and $\Phi_{ion}^o$ are also orthogonal to each other for all interatomic distances, and the electronic character of the wave functions never changes: $\Phi_{HL}^o$ is always neutral and $\Phi_{ion}^o$ is always ionic. Consequently, $\Phi_{HL}^o$ alone cannot describe a bonded molecule because the ionic contribution is completely missing. A correct description of the ground state always needs a linear combination of $\Phi_{HL}^o$ and $\Phi_{ion}^o$. This was demonstrated by McWeeny [48] and discussed by Pilar [49]. Mathematically, the situation is clear: the description of the electronic ground state of $H_2$ needs CI wave functions, either a linear combination of MO CSFs, or a linear combination of VB CSFs made with either orthogonal or non-orthogonal AOs. The three sets of CSFs are different bases for the same two-dimensional state space, two bases are orthogonal and the third one is non-orthogonal. The advantage of the orthogonal VB basis is that the squared CI coefficients have indeed the properties of probabilities and thus are a measure of the ionic character of the state; when non-orthogonal AOs are used, weights of the VB CSFs can only be approximately calculated, for example with the Chirgwin–Coulson formula [50].

The use of orthogonal AOs in VB calculations is not a trivial task, after all, the orthogonality of two basis functions or AOs depends on the molecular geometry. Atomic basis functions or AOs located at the position of atoms in a molecule are in general not orthogonal, but they can be orthogonalized with, e.g., Löwdin's symmetric orthogonalization method, and then used in a VB calculation. Alternatively, one can calculate the electronic state of a molecule with a conventional CI wave function using MO CSFs and then localize the MOs by an orthogonal transformation. I developed a method where delocalized MOs obtained with CASSCF are localized at predefined fragments with the

help of an orthogonal transformation giving orthogonal fragment MOs (FMO). The advantage of this procedure is that the orthogonal transformation in the MO space causes an orthogonal transformation in the CSF space, leaving the CASSCF wave function invariant. Most transformed MOs will be delocalized FMOs but some FMOs resemble atom-centered AOs or hybrid orbitals (HO); these FMOs are called orthogonal AOs (OAO). OAOs include the deformation of the atomic electron distribution due to polarization caused by the molecular environment; in this respect, they are very similar to orthogonalized quasi AOs introduced by Ruedenberg et al. [51,52]. The CASSCF wave function constructed from orthogonal FMOs instead of orthogonal MOs is a linear combination of OVB CSFs with doubly occupied non-active FMOs and active OAOs. The VB-like character of the transformed CSFSCF wave functions is due to the active OAOs. This OVB method [53,54] was used to study the symmetry allowed and forbidden reactions [55]. In this paper, the method is used to analyse the ground state of the $C_2$ molecule, which has $^1\Sigma_g^+$ symmetry. From two carbon atoms in their $^3P_g$ ground states, one can derive three molecular states with *gerade* parity, two are $^1\Sigma_g^+$ states and one is a $^1\Delta_g$ state; from the carbon atoms in the $^5S_u$ state one can derive another $^1\Sigma_g^+$ state. In $D_{\infty h}$ symmetry, $\Sigma$ and $\Delta$ states are automatically orthogonal to each other, but because actual calculations can only be conducted in the largest Abelian subgroup $D_{2h}$, in which $\Sigma_g^+$ and one component of the $\Delta_g$ state are in the same irreducible representation $A_g$, they can mix. The OVB method used in this paper is a tool to analyse CASSCF wave functions; the OVB CSFs are created by the GUGA algorithm implemented in GAMESS, and there is no hand selection of CSFs. Degenerate CSFs are identified using the weights and energies as criteria, and normalized linear combinations are used in the following analysis. Totally symmetric linear combinations of OVB CSFs are labelled LC. Monitoring the weights of the LCs along the whole reaction coordinate allows one to distinguish between neutral and ionic LCs; one can obsevre whether and where certain spin couplings are important, and one can also observe how strongly the electron distribution of atoms in the molecule deviates from the isotropic distribution in the free atoms. In highly symmetric systems such as homonuclear diatomic molecules, the anisotropy is represented by a large number of LCs of different characteristics; most of them are ionic. Moreover, one can compare the electron distribution of excited states with that of the ground state using the same means.

The lowest three of the four $^1A_g$ states of $C_2$ are studied in this paper. To observe how CASSCF wave functions are composed that describe the dissociation of single, double and triple bonds, the molecules ethane, ethene and $N_2$ are also studied.

### 3. Étude: The Description of Single, Double and Triple Bonds

A wave function derived from a closed shell electron configuration is in most cases a single Slater determinant, e.g., the electron configuration $\sigma^2$ with a bonding $\sigma$ MO leads immediately to the MO CSF $|\bar{\sigma}\sigma|$; similarly, when doubly degenerate bonding $\pi$ MOs are fully occupied, the MO CSF $|\bar{\pi}_x\pi_x\bar{\pi}_y\pi_y|$ corresponds to the electron configuration $\pi^4$. If the $\pi$ MOs are not fully occupied, e.g., when the electron configuration is $\sigma^2\pi^2$, two MO CSFs are possible; $|\bar{\sigma}\sigma\bar{\pi}_x\pi_x|$ and $|\bar{\sigma}\sigma\bar{\pi}_y\pi_y|$ and a wave function that has rotational symmetry must be a linear combination of them, and is either $|\bar{\sigma}\sigma\bar{\pi}_x\pi_x| + |\bar{\sigma}\sigma\bar{\pi}_y\pi_y|$ or $|\bar{\sigma}\sigma\bar{\pi}_x\pi_x| - |\bar{\sigma}\sigma\bar{\pi}_y\pi_y|$.

These wave functions are not able to describe dissociation because all MOs are bonding MOs; for a correct description of dissociation, all bonding and the corresponding antibonding MOs must be included into the set of active MOs. The CAS problem is defined by giving the number of active MOs and active electrons, and the order of the active orbitals. Non-active MOs are not explicitly mentioned.

The $2s$ AO and the $2p$ AOs located at atom A will be labelled $s_A$ and $x_A, y_A, z_A$, respectively; analogously, the AOs are on atom B. For HOs, no separate symbol is used; they are labelled by their respective dominant AO. For all molecules discussed, the molecular axis will be the z-axis, and $\sigma$ MOs will be made by HOs; if the HO has a dominantly s-character, the $\sigma$ MO will be labelled $\sigma_s$. A $\sigma_p$ is made with HOs dominated by the $2p_z$ AO.

### 3.1. The Dissociation of the C–C Single Bond in Ethane

The equilibrium C–C distance of ethane is about 1.55 Å; Figure 1 also shows that the orthogonal transformation of the MOs leaves the total energy indeed invariant.

The lowest level wave function that correctly describes the dissociation of the $\sigma$ single bond in ethane is a CAS(2,2) wave function, with $\sigma_p$ and the antibonding $\sigma_p^*$ MO as active MOs and two active electrons. Two frozen core MOs and six MOs describing the CH bonds are doubly occupied, and they are not mentioned in the following.

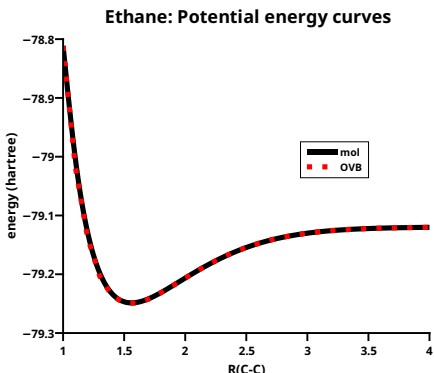

**Figure 1.** The potential energy curves for ethane calculated with MO CASSCF and the OVB method.

The weight curves of the MO CSFs, see Figure 2, demonstrate that the $|\sigma_p^2|$ CSF is, at short distances, a good description of the ethane ground state, but at long distances only a linear combination of $|\sigma_p^2|$ and $|\sigma_p^{*2}|$ can describe the dissociation into two methyl fragments.

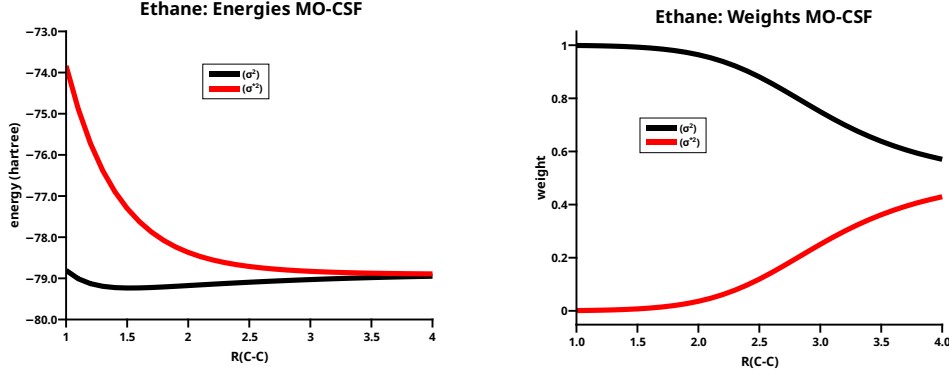

**Figure 2.** Energies (**left**) and weights (**right**) of the two MO CSFs of ethane.

Localization of the 10 MOs onto the two methyl fragments yields two equivalent sets of FMOs, each having one frozen core MO describing the 1s AO, three delocalized FMOs describing the three CH bonds, and one localized FMO having the character of an $sp$-type HO. The eight non-active FMOs, denoted $1_A, 2_A, 3_A, 4_A$ and $1_B, 2_B, 3_B, 4_B$, will not be mentioned. With the two $2p_z$-dominated hybride OAOs, $z_A$ and $z_B$, one can make CSFs, which have the form of the Heitler–London VB wave function, $\Phi_{HL}^o = |(z_A \bar{z}_B - \bar{z}_A z_B)|$, and the ionic CSF $\Phi_{ion}^o = |(z_A \bar{z}_A + z_B \bar{z}_B)|$.

$\Phi_{HL}$ describes the singlet coupling of the doublet states of the methyl groups; each methyl group has one unpaired electron that is ready for bonding and, thus, conforms to the beliefs in chemistry that unpaired electrons are necessary for creating the Lewis electron pair representing a covalent single bond.

The energy curves of $\Phi_{HL}^o$ and $\Phi_{ion}^o$ are completely repulsive, see Figure 3; since McWeeny's early OVB calculations on $H_2$, this is a well known feature of OVB CSFs. The weight curves demonstrate that the neutral CSF $\Phi_{HL}^o$ dominates the geometries at long C–C distances but the ionic CSF $\Phi_{ion}^o$ becomes important at shorter distances when polarization

and interference cause a deviation from the electron distribution of the unperturbed fragments. The ionic CSF $\Phi_{ion}^{o}$ thus describes a shift of the charge distribution in the covalent bond. When the C–C distance goes to zero, the weights of both OVB CSFs become equal. Around the equilibrium geometry, the weight of the neutral CSF $\Phi_{HL}^{o}$ is larger than that of the ionic CSF $\Phi_{ion}^{o}$. A comparison of the weight curves of MO CSFs and OVB CSFs demonstrate antagonistic behaviour: at long distances, where a single OVB CSF correctly describes the dissociated system, a linear combination of $|\sigma^2|$ and $|\sigma^{*2}|$ is necessary to do this; at short distances, where $|\sigma^2|$ is a good approximation to the ground state wave function, a linear combination of $\Phi_{HL}^{o}$ and $\Phi_{ion}^{o}$ is necessary to obtain a qualitative correct ground state wave function. This behaviour is found for all dissociation reactions.

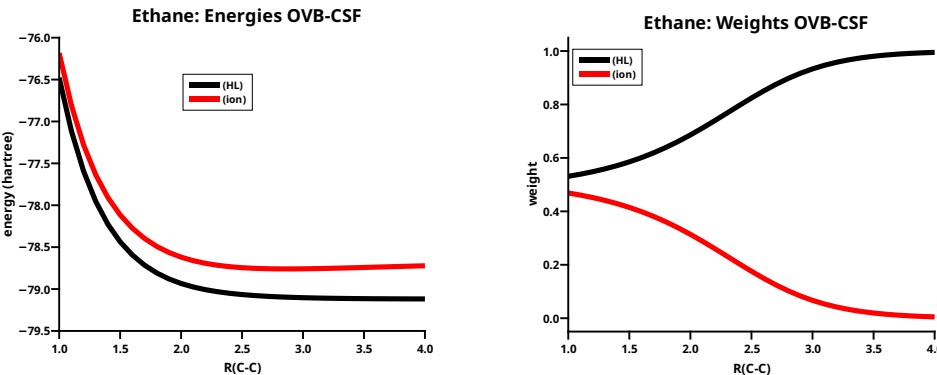

**Figure 3.** Energies (**left**) and weights (**right**) of the two OVB CSFs for ethane.

### 3.2. The Dissociation of the C–C Double Bond in Ethene

The smallest possible wave function that can describe the dissociation of the double bond is the CAS(4,4) wave function with the four active MOs $\sigma$, $\pi$, $\sigma^*$, and $\pi^*$; the corresponding electron configuration is $\sigma^2\pi^2$. The HOs used to make the $\sigma$ MO have a dominantly $z$-character, so they are labelled $z_A$ and $z_B$, and the $\pi$ MOs are made with $x$-OAOs. Figure 4 shows again that the orthogonal transformation of MOs to FMOs leaves the total energy invariant; the equilibrium C=C distance is about 1.35 Å.

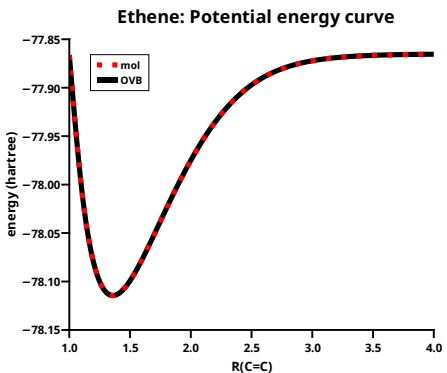

**Figure 4.** The potential energy curves for ethene calculated with MO CASSCF and the OVB method.

The CI space for CAS(4,4) singlet wave functions comprises 20 CSFs, in the $D_{2h}$ symmetry, only eight CSFs are totally symmetric. The order of the four active OAOs in all CSFs is $z_A x_A z_B x_B$. The following notation is used: *aabb* means that one $\alpha$ electron occupies the $z$ OAO and one occupies the $x$ OAO on atom A, and $\beta$ electrons occupy the OAOs on atom B. *a2b0* means: the $z$ OAO of atom A is singly occupied by an $\alpha$ and the $x$ OAO is doubly occupied; the $z$ OAO of atom B is singly occupied by a $\beta$. The OVB CSF *a2b0* does not have $D_{2h}$ symmetry, but the positive linear combination with *a0b2* does.

The 8 totally symmetric linear combinations of OVB CSFs are shown in Table 1.

**Table 1.** The totally symmetric linear combinations of singlet OVB CSFs for ethene.

| LC | CSFs |
|---|---|
| LC1 | $= \lvert aabb \rvert$ |
| LC2 | $= \lvert abba \rvert$ |
| LC3 | $= \lvert 2020 \rvert$ |
| LC4 | $= \lvert 0202 \rvert$ |
| LC5 | $= \lvert 2002 \rvert + \lvert 0220 \rvert$ |
| LC6 | $= \lvert 2a0b \rvert + \lvert 0a2b \rvert$ |
| LC7 | $= \lvert a2b0 \rvert + \lvert a0b2 \rvert$ |
| LC8 | $= \lvert 2200 \rvert + \lvert 0022 \rvert$ |

Carbene is a diradical, and two electrons occupy two carbon-centered, "nearly-degenerate" lone pair HOs, giving rise to a three singlet and one triplet state [56]. Salem and Rowland classified the two states with singly occupied lone pair orbitals as diradical, and the two states with doubly occupied lone pair orbitals as zwitterionic. According to our notation, the $z$ OAO represents the $sp$ HO of methylene, and the $x$ OAO represents the $p$ HO. Using this notation, the four lowest methylene states at the equilibrium geometry, with increasing energy, are $^3(zx)$, $z^2$, $^1(zx)$, and $x^2$.

LC1 to LC5 are neutral, LC6 and LC7 are singly ionic, and LC8 is a doubly ionic LC. LC1 describes the two methylenes in their triplet ground states, coupled to a singlet. The two electrons in the $z$ OAOs form the $\sigma$ bond, the two electrons in the $x$ OAOs form the $\pi$ bond, and LC1 is nothing but the Heitler–London portion of the $\sigma$ and the $\pi$ bonds. Since the methylenes are in high spin states, the unpaired electrons are "ready for bonding". In the dissociated molecular system, LC1 describes two noninteracting methylenes in their respective electronic ground states; at all other geometries, the methylenes are no longer in a methylene eigenstates, because interacting subsystems of a system are never in pure states but always in mixed states [57,58]. In these cases, LC1 describes two "local triplet states", that is, "local high-spin states", coupled to a singlet. This is, what the Heitler–London CSF represents. That "local low-spins states" are rather unimportant for bonding shows LC2, where each methylene is in the singlet diradical state, which is considerably higher than the triplet diradical state. Moreover, the spins are not unpaired and therefore not "ready for bonding" although the same AOs are singly occupied as in the case of LC1. LC3 describes two methylenes both with doubly occupied $z$ OAOs, and LC4 describes two methylenes with doubly occupied $x$ OAO; in both CSFs, the active electrons are singlet coupled and therefore not "ready for bonding", the contributions of these CSFs to the ground state wave function are accordingly very small. The singly ionic LC6 describes the shift of one electron in the $\sigma$ bond, and the $\pi$ MO is doubly occupied; the singly ionic LC7 describes the shift in the $\pi$ bond with doubly occupied $\sigma$ MO. These two singly ionic LCs are necessary to describe polarization in the $\sigma$ and the $\pi$ orbital, respectively. Without them, covalent bonding cannot be correctly described. The neutral LC5 describes local angular correlation: If atom A is in the low lying zwitterionic methylene state, atom B is in the high lying zwitterionic state. This is the fourth LC that contributes significantly to the ground state of ethene. LC8 describes dianion/dication pairs.

Figure 5 shows energies and weights of the four large LCs, which are LCs having a weight larger than 0.1 somewhere along the reaction coordinate. All other LCs are small LCs. LC1 has the lowest energy along the whole reaction coordinate; this demonstrates the importance of the coupling local high-spin states to a global low spin state. Although the ionic LCs LC6 and LC7 have nearly identical energies, their weights are very different. The weight of LC6 reaches a value of 0.1 at a C–C distance of 2.7 Å, but that of LC7 only at 1.8 Å. LC5 becomes important only when the two carbon atoms are rather close; the weight is larger than 0.1 only at C–C distances shorter than the equilibrium distance. Note that LC6 and LC7 have higher energies than LC5 but their weights are much larger. LC6 and LC7 demonstrate the importance of the charge shift for covalent bonding, and LC5

demonstrates that angular correlation becomes important in multiple bonds as soon as the interacting atoms come close. It is noteworthy that, at the equilibrium distance, the weight of the neutral LC1 is 0.32, which is only slightly larger than the weight of the ionic LC6 (0.29), whereas LC7 has a weight of only 0.17; nevertheless, the sum of the weights of the ionic LCs is much larger than the sum of the two neutral ones. At C–C distances longer than 3.5 Å, the weight of LC1 is 1.; between 3.5 and 2.7 Å the weight of LC1 decreases and that of LC6 increases, but the sum of both LCs is still close to 1. Then, LC7 and LC5 become gradually more important, but the sum of all four large LCs nevertheless decreases down to 0.85 at a C–C distance of 1.0 Å. At the same time, the weight of the four small LCs increases to 0.15, see Figure 6; nevertheless, at the equilibrium distance, the sum of their weights is only 0.11.

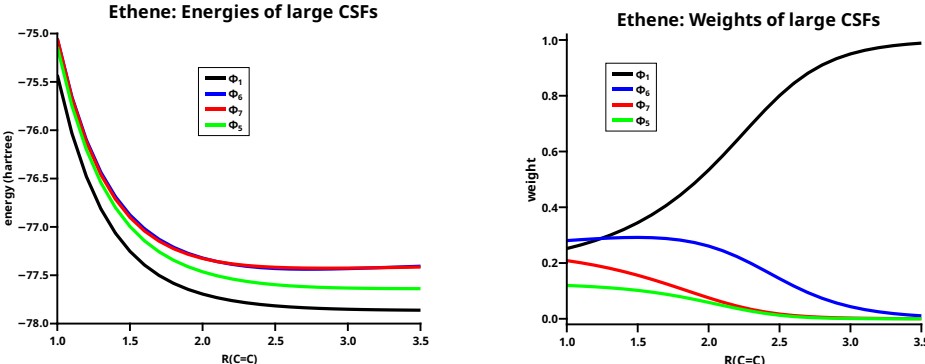

**Figure 5.** Energies (**left**) and weights (**right**) of the large LCs for ethene.

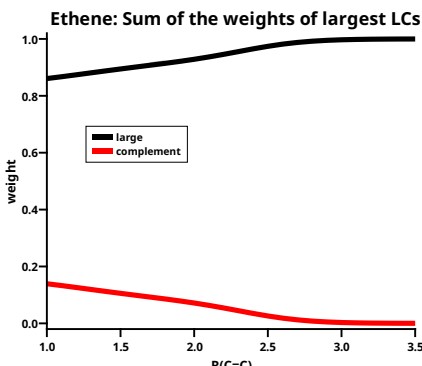

**Figure 6.** The sum of the weights of the large LCs for ethene.

### 3.3. The Dissociation of the N–N Triple Bond in the Nitrogen Molecule

The electron configuration of the nitrogen molecule is $\sigma_p^2 \pi^4$, where $\sigma_p$ is the bonding linear combination of $z_A$ and $z_B$; $\pi^4$ means $\pi_x^2 \pi_y^2$, $\pi_x$ and $\pi_y$ are bonding linear combinations of the respective OAOs.

Figure 7 shows the potential energy curves obtained with MO CSFs and OVB CSFs; the equilibrium distance of N$_2$ is 1.1 Å.

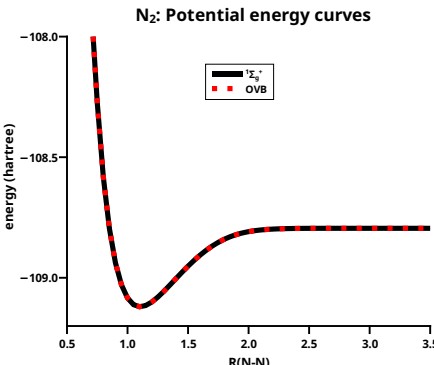

**Figure 7.** The potential energy curves for $N_2$ calculated with MO CASSCF and the OVB method.

The dissociation reaction in the lowest $^1\Sigma_g^+$ state is correctly described by a CAS(6,6) wave function with the six valence MOs $\sigma_p$, $\pi_x$ and $\pi_y$ and the corresponding antibonding MOs; the $2s$ AOs are always doubly occupied. In $D_{2h}$, there are 32 totally symmetric singlet CSFs, 20 of them are not zero. The OVB calculation is conducted in $C_{2v}$; from the 55 OVB CSFs only, 21 linear combinations are not zero, and they are shown in Table 2. The order of the six OAOs in the CSFs is $y_A, x_A, z_A, y_B, x_B, z_B$. From the 21 LCs, LC1 to LC9 are neutral, LC10 to LC18 are singly ionic, LC19 and LC20 are doubly ionic, and LC21 is triply ionic.

**Table 2.** The totally symmetric linear combinations of singlet CSFs for $N_2$.

| LC | | CSFs |
|---|---|---|
| LC1 | $=$ | $\|aaabbb\|$ |
| LC2 | $=$ | $\|ababab\|$ |
| LC3 | $=$ | $\|baaabb\|$ |
| LC4 | $=$ | $\|baabab\| + \|abaabb\|$ |
| LC5 | $=$ | $\|a02b20\| - \|0a22b0\| - \|2a00b2\| + \|a20b02\|$ |
| LC6 | $=$ | $\|0a20b2\| + \|a02b02\|$ |
| LC7 | $=$ | $\|02a02b\| + \|20a20b\|$ |
| LC8 | $=$ | $\|20a02b\| + \|02a20b\|$ |
| LC9 | $=$ | $\|a20b20\| + \|2a02b0\|$ |
| LC10 | $=$ | $\|202002\| + \|022002\| + \|002022\| + \|002202\|$ |
| LC11 | $=$ | $\|2ba0ab\| - \|b2aa0b\| - \|b0aa2b\| + \|0ba2ab\|$ |
| LC12 | $=$ | $\|2aa0bb\| - \|a2ab0b\| - \|a0ab2b\| + \|0aa2bb\|$ |
| LC13 | $=$ | $\|220020\| + \|220200\| + \|200220\| + \|020220\|$ |
| LC14 | $=$ | $\|202020\| + \|022200\| + \|200022\| + \|020202\|$ |
| LC15 | $=$ | $\|022020\| + \|202200\| + \|020022\| + \|200202\|$ |
| LC16 | $=$ | $\|aa2bb0\| + \|aa0bb2\|$ |
| LC17 | $=$ | $\|ab2ba0\| + \|ab0ba2\|$ |
| LC18 | $=$ | $\|002220\| + \|220002\|$ |
| LC19 | $=$ | $\|22a00b\| + \|00a22b\|$ |
| LC20 | $=$ | $\|2a20b0\| - \|a22b00\| - \|a00b22\| + \|0a02b2\|$ |
| LC21 | $=$ | $\|222000\| + \|000222\|$ |

From these LCs only five are large: LC1, LC5, LC12, LC16, and LC20. See left side of Figure 8.

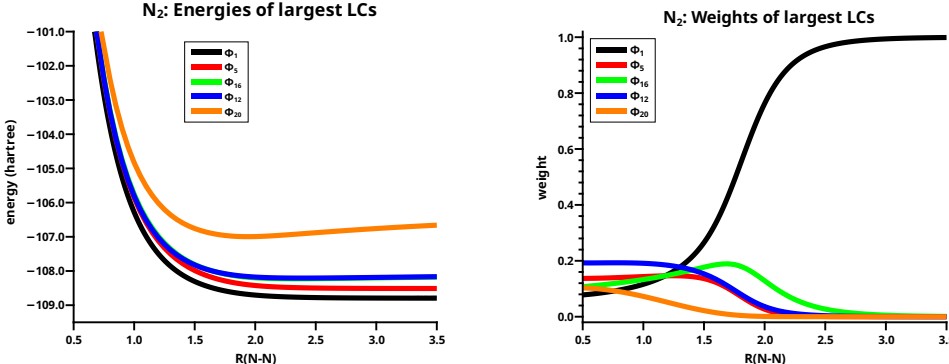

**Figure 8.** Energies (**left**) and weights (**right**) for $N_2$ of the five large LCs.

The neutral LC1 describes the singlet coupling of the high spin quartet states; it is the Heitler–London contribution to the triple bond in $N_2$; the neutral LC5 describes the angular correlation and a spin flip from the local quartet state into the low-spin doublet state; the singly ionic LCs LC16 and LC20 describe the shift of electrons in the $\sigma$ and in the $\pi$ MOs, respectively. The doubly ionic LC20 describes the simultaneous shift of an electron in the $\sigma$ and an electron in the $\pi$ MOs. LC1, LC16 and LC20 are of major importance for the description of the triple bond in $N_2$, the two ionic LCs have, as found for the ethene molecule, nearly identical energies, but the weights are rather different. LC16 becomes important at an N–N distance of about 2.7 Å, and LC20 at a distance of about 2.2 Å. The reason for this are the different spatial extensions of the involved AO; the $z$ AOs, which are aligned along the molecular axis, interfere earlier than the perpendicular $x$ and $y$ AOs, and therefore the charge shift in the $\sigma$ bond starts earlier than in the $\pi$ bonds.

Ten of the remaining 16 LCs are small and six are effectively zero. The weights of all small LCs, shown in the left side of Figure 9, are effectively zero for N–N distances longer than 2.5 Å, but, as for the large ionic LCs, most of them contribute significantly at N–N distances smaller than 2.0 Å. The right side of Figure 9 shows that the sum of the weights of the ten small LCs (labelled complement in the legend) around the equilibrium distance is about 30%. All small LCs are important in describing the deviation of the atomic charges from the spherical symmetry during bonding. LC2, LC3, and LC8 are neutral, LC19 and LC21 are doubly and triply ionic, respectively, and all others are singly ionic.

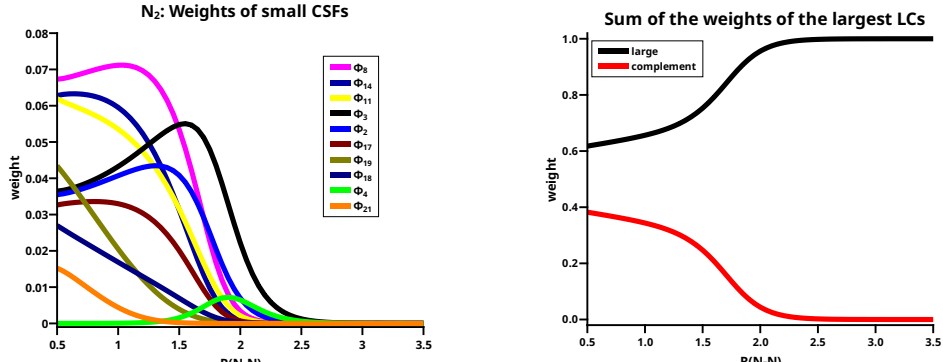

**Figure 9.** (**Left**): Weights of the small LCs for $N_2$. (**Right**): Sum of the weights of the large and the small LCs.

## 4. Bonding in $C_2$

The equilibrium distance in the singlet ground state is close to the grid point $R = 1.25$ Å. If one assumes that the electron configuration of $C_2$ is similar to that in $N_2$, it must be $\sigma_s^2 \sigma_s^{*2} \pi^4$; the bonding $\sigma_s$ and the antibonding $\sigma_s^*$ MOs spanned by the 2s AOs are doubly occupied and the remaining four valence electrons occupy the two bonding $\pi$

MOs. All four MOs are non-active and the wave function is a single Slater determinant. This wave function is not able to describe dissociation; the most simple wave function that can do this is a CAS(4,4) wave function with four active MOs, two bonding and two antibonding $\pi$ MOs, and four active $\pi$ electrons. If one considers that the bonding $\sigma_p$ MO has a lower energy than the bonding $\pi$ MOs, one obtains a second possible electron configuration: $\sigma_s^2 \sigma_s^{*2} \sigma_p^2 \pi^2$. The $\sigma_p$ MO is non-active but the doubly degenerate $\pi$ MOs are occupied by only two electrons, and thus active MOs. The wave functions corresponding to this electron configuration are CAS(2,2) wave functions, which cannot describe disso­ciation, because they contain no antibonding MOs, but CAS(4,6) wave functions with all six MOs spanned by $2p$ AOs as active MOs can do it. In Figure 10, one can observe that the stabilization of the ground state as calculated with both CAS(4,4) and CAS(4,6) are far too low, and the equilibrium distance obtained with CAS(4,6) is considerably longer than that obtained with CAS(4,4). The long equilibrium distance stems from the fact that with CAS(4,6), the ground state is a $^1\Delta_g$, whereas with CAS(4,4) it is a $^1\Sigma_g^+$ state; however, the poor stabilization indicates that wave functions without active $\sigma_s$ and $\sigma_s^*$ MOs cannot describe the ground state correctly. The two $\sigma_s$ MOs and the four electrons must become active. Then, one has eight active MOs and eight active electrons; with such a CAS(8,8) wave function, the lowest singlet state is indeed the ground state of $C_2$, and the stabilization energy is reliable.

The $\sigma_s$ MOs are spanned by hybride AOs that are nearly pure $s$ AOs; thus, the fact that lobes point inwards in the $\sigma_s$ MO and point outwards in the $\sigma_s^*$ MO is no surprise. The fact that in the second pair of $\sigma$ MOS it is the other way around is also not surprising. These MOs are spanned by hybride AOs with a large $2p_z$ contribution. The two bonding and the two antibonding $\sigma$ MOs are optimized in two-dimensional subspaces of the space of active MO; each of the energetic low lying $\sigma_s$ MOs specifies a spatial electron distribution, and the higher lying $\sigma_z$ MOs must accomodate accordingly. Therefore, the large lobes of the hybride AOs in $\sigma_z$ are pointing outwards, and in the $\sigma_z^*$ MO they are pointing inwards.

The shape of the ground state PEC, as calculated with the CAS(8,8) wave function, indicates the avoided crossings that are not found with the two smaller CAS wave functions. See left side of Figure 10.

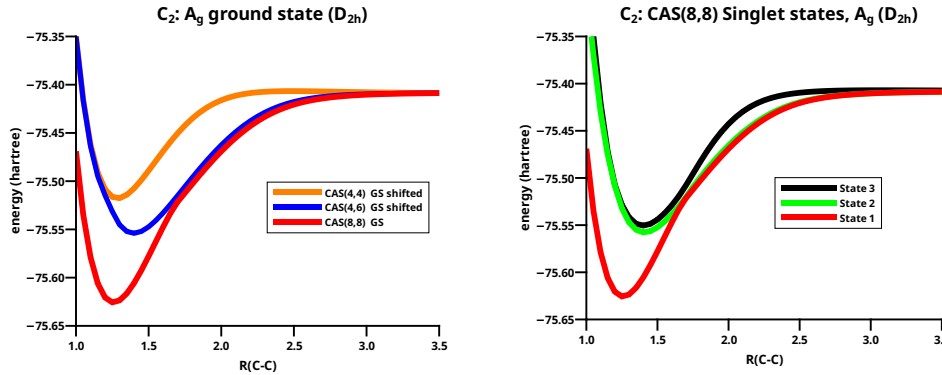

**Figure 10.** (**Left**): Potential energy curves of the lowest $^1A_g$ states for $C_2$ calculated with CAS(4,4), CAS(4,6), and CAS(8,8) wave functions. (**Right**): The CAS(8,8) potential energy curves for the three lowest $^1A_g$ states.

As mentioned above, the avoided crossings are the result of making all CASSCF calculations in the $D_{2h}$ symmetry. The totally symmetric CAS(8,8) singlet wave function of $A_g$ symmetry is a linear combination of 264 CSFs. The ground state wave function is indeed a mixture of $^1\Sigma_g^+$ and $^1\Delta_g$ states, which results in avoided crossings, as can be observed for the three lowest singlet states investigated. (Details of the calculations with MO CSFs can be found in the Supporting Informations.) The PECs of the three $A_g$ states agree very well with those reported by Boschen et al. [59]. That means that the wave functions for the states have a different character in certain regions, e.g., the ground state has a $\Sigma_g^+$ character

around the equilibrium geometry, the second $A_g$ state has a $\Delta_g$ character, and the third $A_g$ state has again a $\Sigma_g^+$ character; the second and the third state are energetically very similar and also the local minima have similar geometries. See the right side of Figure 10.

The ground state has a $\Sigma_g^+$ character for C–C distances shorter than 1.70 Å; there, the CSF derived from the electron configuration $\sigma_s^2 \sigma_s^* \pi_x^2 \pi_y^2$ contributes about 70% to the wave function, and the CSF derived from $\sigma_s^2 \sigma_p^2 \pi_x^2 \pi_y^2$ contributes about 10%. Indeed, at an interatomic distance of 1.25 Å, it is 71% and 13%, in agreement with the numbers reported by Hermann and Frenking [33]. Ignoring all CSFs that contribute in total 16% to the wave function, what bond order has a system with 71% bond order 2 and 13% bond order 4? In my opinion, what is more important than answering this question is to find out how spins and charges rearrange on the way from two isolated atoms to the molecule; that is, how the interaction of the atoms disturbs their electron distributions during the recombination reaction, or how the electron and spin arrangement in the molecule readjust to that in the free atoms during dissociation.

*OVB Analysis of Bonding in $C_2$*

Every atom in a homonuclear diatomic molecule has a $C_{\infty v}$ symmetry, but all actual calculations are made in the Abelian subgroup $C_{2v}$ of $D_{2h}$. With the eight OAOs $s_A, z_A, x_A, y_A$, $s_B, z_B, x_B, y_B$, 492 singlet OVB CSFs can be made, and only very few OVB CSFs have already $g$ parity or have rotational symmetry; more often than not, only LCs have it. This means that a large number of OVB CSFs have zero weight because of symmetry reasons. However, LCs can also gain zero weight when the molecule's geometry changes, and the number of LCs with non-zero weight depends strongly on the geometry; it is nevertheless rather large. As a consequence, many small LCs can make considerable contributions, and if only large LCs are considered, the description of the wave functions is not satisfying because the weights of many large LCs can be rather small at certain geometries. To consider also small LCs that make, nevertheless, large contributions to the wave functions, one must define LCs as significant if they have weights larger than 0.01 somewhere along the reaction coordinate. A total of 128 significant LCs are found to describe the lowest three $A_g$ singlet states along the whole reaction coordinate; in detail, 51 significant LCs are found to contribute to the first $^1A_1$ state (the ground state), 63 LCs to the second state, and 80 LCs to the third singlet state; only 15 LCs of the significant LCs are large. From these LCs, ten contribute to the description of the first $^1A_1$ state, and eight LCs contribute to the second and to the third state, respectively. Details may be found in the Supporting Information.

Table 3 lists the ten large LCs found for state 1 along the reaction coordinate, in the left side of Figure 11 the weight curves are shown. Starting at long C–C distances, one can observe that only two LCs are important, LC05 ($\approx$75%t) and LC06 ($\approx$20%). Both LCs describe the singlet coupling of two carbon atoms in their respective $^3P_g$ ground states; in LC06, the electrons are located in the $x$ and the $y$ OAO, thus LC06 describes the formation of two $\pi$ bonds. The situation where in each atom one electron occupies the $z$ and the other either the $x$ OAO or the $y$ OAO is described by LC05. Singlet coupling gives then either a $\sigma_p$ and a $\pi_x$ bond or a $\sigma_p$ and a $\pi_y$ bond. The positive linear combination of these two LCs has $\Sigma_g^+$ symmetry and is represented by LC05; the negative linear combination has $\Delta_g$ symmetry and is represented by LC01. Both $2s$ OAOs are always doubly occupied. At very long C–C distances, the two carbons atoms are completely interaction-free, and the three ways of distributing two spins in three $p$ orbitals are equivalent, and the weight of LC05 is 2/3, and that of LC06 is 1/3, but when the atoms approach each other, the interaction along the molecule axis becomes more favorable so that the weight of LC05 increases. At a C–C distance of 3.5 Å, the weights of LC05 and LC06 are indeed 74% and 21%, respectively; at 3.0 Å the weights are 78% and 14%, respectively. At these distances, the missing LCs that describe either the polarization in the direction of the molecular axis or the superposition of the $2p_z$ orbitals are not represented by large but only by significant LCs.

At C–C distances less than 3.0 Å, linear combinations LC01 to LC04 with a $\Delta$ character dominate the ground state. LC01 and LC03 describe neutral atomic charge distributions,

LC02 and LC04 describe cation/anion pairs. LC01 describes a $\sigma_p$ and a $\pi$ bond, it is the counterpart of LC05 with the different phase. LC01 describes the neutral Heitler–London component of the C–C $\sigma$ bond and $\pi$ bond; LC02 ionic component of the $\sigma$ bond. The ionic LC04 describes deformations of the interacting C atoms caused by polarization in the $\sigma$ bond due to *sp* hybridization. Polarization in the $\sigma$ bond is also described by the neutral LC03. For C–C distances less than 1.7 Å, the wave function is again dominated by LCs with the $\Sigma_g^+$ character; the large LCs that are important at long distances contribute very little, and LC05 is essentially vanished. The LC06 goes to zero rapidly; the wave function is dominantly a superposition of small LCs that represent the deformation of the electron distribution of the C atoms due to polarization, interference, angular correlation and so on. Around the equilibrium geometry the LC07 gains weight; this neutral LC with an $\Sigma_g^+$ character represents both C atoms in quintet high spin states coupled to a singlet. The quintet state is the result of the excitation of an electron from the doubly occupied 2*s* AO into the 2*p* subshell together with a spin flip. This LC has between 1.65 Å and 1.0 Å a rather constant weight of about 10%. Formally, one could say LC07 represents a quadruple bond which becomes important around the C–C equilibrium distance. LC08 is an ionic LC; it describes the polarisation of all four formal bonds as described by LC07; LC09 is a neutral LC that can be best described as an angular correlation in the two $\pi$ bonds as described by LC06, whereas LC10 describes the polarization in the $\sigma$ bond. The sum of weights of the large LCs decreases dramatically with the C–C distance approaches the equilibrium value, at the same time, the contribution of the small LCs becomes large. This is shown on the right side of Figure 11. Only if the criterion for "being large" is reduced to 0.03, the "large" LCs contribute more than 50% along the whole reaction coordinate, and it is for short C–C distances where these LCs contribute most.

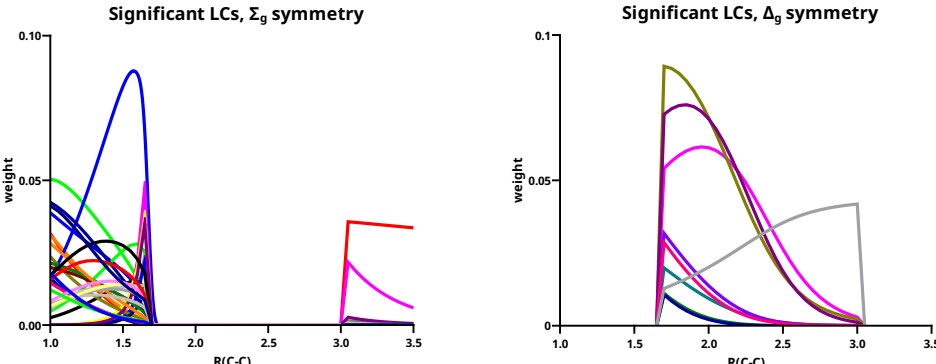

**Figure 11.** (**Left**): The distribution of the weights of the large LCs for the first $A_1$ singlet state of $C_2$. (**Right**): The sum of the large weights and the complement.

Figure 12 shows the increase in the contributions of small LCs with the decreasing C–C distance. Because there are so many of them, the curves are not labelled.

The second $A_1$ state changes its character three times: starting from short C–C distances, it changes at 1.2 Å from the $\Sigma_g^+$ to $\Delta_g$ character, at 1.7 Å from $\Delta_g$ to $\Sigma_g^+$ and at 3.0 Å again to the $\Delta_g$ character. This characterization is due to the eight large LC contributions that dominate the second $A_1$ singlet state; six of them also contribute to the first $A_1$ state. See Table 4 and the left side of Figure 13. The dissociated molecule is solely described by LC01, the molecule with two $\pi$ bonds has always $\Sigma$ symmetry. In the region between 1.7 Å and 3.0 Å LC05 and LC06 dominate the wave function; however, as mentioned above, because of the interaction between the atoms, one $\sigma$ and one $\pi$ bond are more stable than two $\pi$ bonds and therefore LC05 has a much larger weight than LC06. In this region, the two singly ionic LCs LC11 and LC12 describe the polarization in the $\sigma$ bond. The weight of the LC describing the singlet coupled quintet states is well below 0.1 in the second $A_1$ singlet state. The right side of Figure fig:CAS88St2largeLCs shows that below 1.7 Å the sum of the weights of the large LCs is much smaller than that of the small LCs.

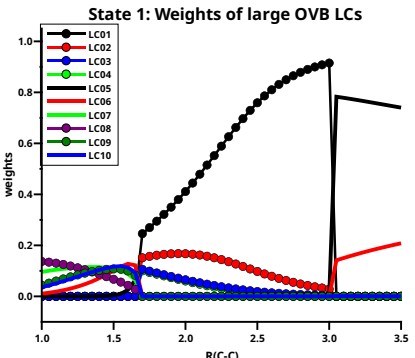
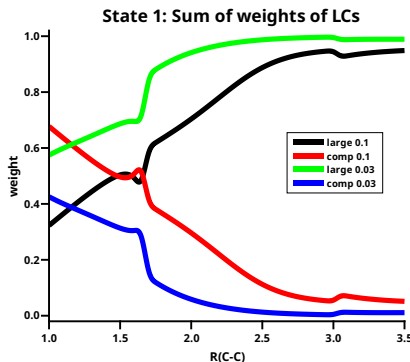

**Figure 12.** (**Left**): The weights of the significant $\Sigma_g$ LCs for the first $A_1$ singlet state of $C_2$. (**Right**): The weights of the significant $\Delta_g$ LCs for the first $A_1$ singlet state.

**Table 3.** Description of the large LCs in the first $A_1$ singlet state of $C_2$. Col 2: IRREPs in $D_{\infty h}$. Col 3: Neutral (n) or singly ionic (s) LC. Col 4: AO symbols. Without normalization coefficients.

| | | | |
|---|---|---|---|
| LC01 | $\Delta_g$ | n | $s_A^2 s_B^2 z_A \bar{z}_B (x_A \bar{x}_B - y_A \bar{y}_B)$ |
| LC02 | $\Delta_g$ | s | $s_A^2 s_B^2 (z_A^2 + z_B^2)(x_A \bar{x}_B - y_A \bar{y}_B)$ |
| LC03 | $\Delta_g$ | n | $(s_A^2 z_A s_B z_B^2 - z_A^2 s_B^2 s_A \bar{z}_B)(x_A \bar{x}_B - y_A \bar{y}_B)$ |
| LC04 | $\Delta_g$ | s | $(s_A^2 z_A^2 \bar{s}_B z_B - s_B^2 z_B^2 s_A \bar{z}_A)(x_A \bar{x}_B - y_A \bar{y}_B)$ |
| LC05 | $\Sigma_g^+$ | n | $s_A^2 s_B^2 z_A \bar{z}_B (x_A \bar{x}_B + y_A \bar{y}_B)$ |
| LC06 | $\Sigma_g^+$ | n | $s_A^2 s_B^2 x_A \bar{x}_B y_A \bar{y}_B$ |
| LC07 | $\Sigma_g^+$ | n | $s_A \bar{s}_B z_A \bar{z}_B x_A \bar{x}_B y_A \bar{y}_B$ |
| LC08 | $\Delta_g$ | s | $((y_A^2 + y_B^2)x_A \bar{x}_B - (x_A^2 + x_B^2)y_A \bar{y}_B)s_A \bar{s}_B z_A \bar{z}_B$ |
| LC09 | $\Delta_g$ | n | $s_A^2 z_A \bar{s}_B (x_B^2 y_A \bar{y}_B - y_B^2 x_A \bar{x}_B) + s_B^2 s_A \bar{z}_B (y_A^2 x_A \bar{x}_B - x_A^2 y_A \bar{y}_B)$ |
| LC10 | $\Sigma_g^+$ | s | $(s_A^2 z_A \bar{s}_B - s_B^2 s_A \bar{z}_B)x_A \bar{x}_B y_A \bar{y}_B$ |

Table 5 shows the 8 large LCs describing the third $A_1$ singlet state, the weight curves are shown in the left side of Figure 14. The third $A_1$ singlet state has an $\Sigma_g^+$ character along the whole reaction coordinate, the dissociated molecule is, like the first state, described by LC05 and LC06 but now the ratio of the weights is 1:2. When the atoms approach the character of the state changes smoothly, the weights of LCs LC05 and LC06 decrease at C–C distance shorter then 2.5 Å where LC15 becomes more important. At 1.7 Å there is again a change of the dominant LCs and again at 1.2 Å. The sum of the large weights becomes nevertheless very small, and the small LCs make the largest contribution. See the right side of Figure 14.

**Table 4.** Description of the large LCs in the second $A_1$ singlet state of $C_2$. Col 2: IRREPs in $D_{\infty h}$. Col 3: Neutral (n) or singly ionic (s) LC. Col 4: AO symbols.

| | | | |
|---|---|---|---|
| LC01 | $\Delta_g$ | n | $s_A^2 s_B^2 z_A \bar{z}_B (x_A \bar{x}_B - y_A \bar{y}_B)$ |
| LC02 | $\Delta_g$ | s | $s_A^2 s_B^2 (z_A^2 + z_B^2)(x_A \bar{x}_B - y_A \bar{y}_B)$ |
| LC03 | $\Delta_g$ | n | $(s_A^2 z_A s_B z_B^2 - z_A^2 s_B^2 s_A \bar{z}_B)(x_A \bar{x}_B - y_A \bar{y}_B)$ |
| LC04 | $\Delta_g$ | s | $(s_A^2 z_A^2 \bar{s}_B z_B - s_B^2 z_B^2 s_A \bar{z}_A)(x_A \bar{x}_B - y_A \bar{y}_B)$ |
| LC05 | $\Sigma_g^+$ | n | $s_A^2 s_B^2 z_A \bar{z}_B (x_A \bar{x}_B + y_A \bar{y}_B)$ |
| LC06 | $\Sigma_g^+$ | n | $s_A^2 s_B^2 x_A \bar{x}_B y_A \bar{y}_B$ |
| LC11 | $\Sigma_g^+$ | s | $(s_A^2 z_A^2 \bar{s}_B z_B - s_B^2 z_B^2 s_A \bar{z}_A)(x_A \bar{x}_B + y_A \bar{y}_B)$ |
| LC12 | $\Sigma_g^+$ | n | $s_A^2 s_B^2 (z_A^2 + z_B^2)(x_A \bar{x}_B + y_A \bar{y}_B)$ |

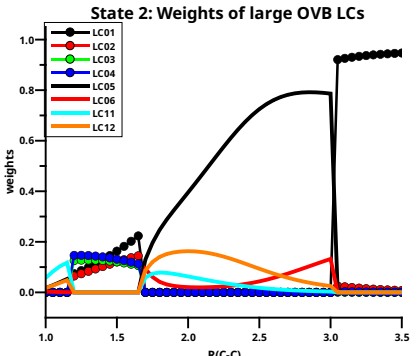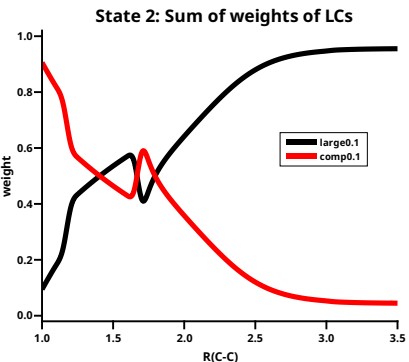

**Figure 13.** (**Left**): The distribution of the weights of the large LCs for the second $A_1$ singlet state of $C_2$. (**Right**): The sum of the large weights and the complement.

**Table 5.** Description of the large LCs in the third $A_1$ singlet state of $C_2$. Col 2: IRREPs in $D_{\infty h}$. Col 3: Neutral (n) or singly ionic(s) LC. Col 4: AO symbols.

| | | | |
|---|---|---|---|
| LC05 | $\Sigma_g^+$ | n | $s_A^2 s_B^2 z_A \bar{z}_B (x_A \bar{x}_B + y_A \bar{y}_B)$ |
| LC06 | $\Sigma_g^+$ | n | $s_A^2 s_B^2 x_A \bar{x}_B y_A \bar{y}_B$ |
| LC13 | $\Sigma_g^+$ | n | $(s_A^2 z_A^2 \bar{s}_B z_B + s_A \bar{z}_A s_B^2 z_B^2)(x_A \bar{x}_B + y_A \bar{y}_B)$ |
| LC14 | $\Sigma_g^+$ | s | $(s_A^2 z_B^2 z_A \bar{s}_B + s_A \bar{z}_B z_A^2 s_B^2)(x_A \bar{x}_B + y_A \bar{y}_B)$ |
| LC15 | $\Sigma_g^+$ | n | $(s_A^2 z_A s_B + s_B^2 s_A \bar{z}_B) x_A y_A \bar{x}_B \bar{y}_B$ |
| LC16 | $\Sigma_g^+$ | s | $(s_A^2 z_A^2 \bar{s}_B z_B - s_B^2 z_B^2 s_A \bar{z}_A)(x_A \bar{x}_B + y_A \bar{y}_B)$ |
| LC17 | $\Sigma_g^+$ | n | $(s_A^2 z_B^2 z_A \bar{s}_B - s_A \bar{z}_B z_A^2 s_B^2)(x_A \bar{x}_B + y_A \bar{y}_B)$ |
| LC18 | $\Sigma_g^+$ | s | $s_A^2 s_B^2 (z_A^2 + z_B^2)(x_A \bar{x}_B + y_A \bar{y}_B)$ |

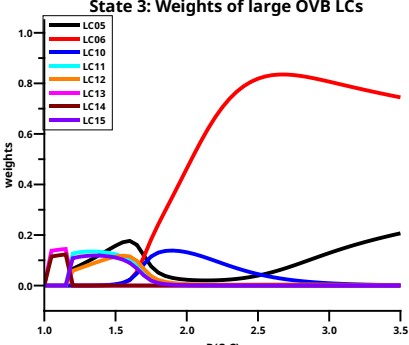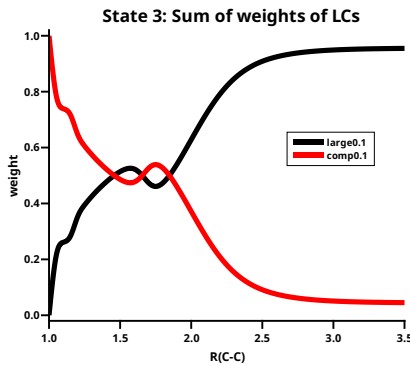

**Figure 14.** (**Left**): The distribution of the weights of the large LCs for the third $A_1$ singlet state of $C_2$. (**Right**): The sum of the large weights and the complement.

## 5. Discussion

The chemical bonding of molecular fragments is accompanied by a reduction of the total electronic energy when the bonded atoms are near to each other. Thereby, the spatial region is enlarged in which bonding electrons can reside. Covalent chemical bonding is of a purely quantum theoretical origin; it is the result of constructive interference when states of the interacting fragments are superimposed, meaning that the probability for finding the shared bonding electrons between the interacting fragments is higher than the sum of the probabilities calculated with the wave functions of the non-interacting fragments. This can also be interpreted as a charge shift, which causes a deformation of the fragment's charge distributions together with classical interactions such as the Coulomb attraction and repulsion of electrons and nuclei. This was demonstrated to be responsible for the stabilization of one electron system such as $H_2^+$, which means that the energetic stabilization is a 1-electron effect but not a 2-electron effect, as suggested by the important role of the Lewis electron pair. In many-electron systems, the fermionic character of the

electrons becomes of the utmost importance for the deformation of charge distributions due to the tendency of identical fermions to avoid coming spatially close, as expressed by the PEP, which was expressed by Lévy-Leblond and Balibar in the following way: *A system of fermions can never occupy a configuration of individual states in which two individual states are identical* [60]. This tendency is so important because it is independent of physical properties such as the electric charge, after all, it holds also for protons and neutrons in nuclei where nuclear forces act. In chemistry, individual states of electrons are called spinorbitals; they can be localized AOs but also delocalized MOs. That two identical electrons can never be found in the same place (Fermi correlation) becomes clear only if eigenstates of the position or localization operator are considered as individual states. However, this holds only if the electrons have identical properties, including the spin projection. Mathematically, the antisymmetry of the state function of a many-fermion system expresses this tendency, and it ... *plays the role of a fictitious, although highly effective, mutual repulsion being exerted within the system, irrespective of any other actual forces or interactions [. . . ] that might be present* [60]. Using loose language, one speaks of Pauli repulsion, which keeps identical electrons apart, thereby reducing the Coulomb repulsion. This tendency is not restricted to electrons in an atom or in a molecule but it is operative also between atoms or molecules when they come close, for example, in condensed matter or during chemical reactions. The PEP explains the shell structure of many-electron atoms, but also the origin of certain bond angles in molecules for which mainly the valence electrons are responsible. Before continuing with the role of the PEP, there is an important caveat: Electrons in atoms or molecules cannot be individualized; one says they are indistinguishable, and this means that it is not possible to attribute a certain individual state to each of them. One can only speak of a configuration of individual states and say that all electrons together occupy these individual states. Nevertheless, again using loose language, one says that a certain electron is in a certain state or a certain electron has certain properties. In the following, I will also use this simple way to speak about a complex issue. The valence electrons in an atom occupy a spherical shell with a characteristic radius and thickness; the radius of the spherical shell is approximately equal to the maximum of the radial density of the valence AOs. For atoms in the second row of the periodic table, the radial densities of the $2s$ and the $2p$ AOs are nearly identical; this is not true for all higher rows. Thus, $2s$ electrons and $2p$ electrons reside in the same spatial area irrespective of the different orbital energies, and, according to the PEP, the electrons with identical spin will prefer relative positions with a maximum distance to all others. Two identical electrons will prefer to be on different sides of the nucleus; this means in an electron–nucleus–electron angle of 180 degrees, three identical electrons will prefer a trigonal arrangement with three angles of about 120 degrees, and four identical electrons will prefer a tetrahedral arrangement [61]. In a noble gas, the valence shell is occupied by eight electrons, four of which are identical $\alpha$ electrons and four are identical $\beta$ electrons; therefore, for both groups of identical electrons, the probability for tetrahedral spatial arrangements will be the highest of all possible. Coulomb repulsion maximizes the distance between $\alpha$ and $\beta$ electrons (Coulomb correlation), giving two interpenetrating tetrahedra inscribed into a cube. This was called a "cubical atom" by Lewis [3]; that such arrangements can be found in many-electron atoms was demonstrated by Scemema et al. [62] using correlated electron structure methods. As soon as the free atom is disturbed, as it is in a chemical reaction, the electron distribution changes. Starting from the noble gas electron configuration in, say, the fluoride anion $F^-$, the creation of an F-H single bond by the interaction with a proton can be observed as the rearrangement of the two tetrahedra when a proton approaches the $F^-$ and attracts electrons in the valence shell. One can assume that the electron at the corner of one tetrahedron, say of the $\alpha$ electrons, will be attracted and the tetrahedron will rearrange so that the corner points towards the proton. However, the proton can attract another electron, but this must be a $\beta$ electron; the Coulomb repulsion of the two electrons close to the proton is much smaller than the reluctance of two $\alpha$ electrons coming close. This causes a reorientation of the two tetrahedra bringing two corners in approximate coincidence; the two electrons are the bonding electron pair. The other six

electrons can be thought of forming a regular hexagon with alternating $\alpha$ and $\beta$ electrons at the corners. Starting from the $O^{2-}$ dianion, one can add stepwise two protons by which eventually all four corners of the two tetrahedra are brought into approximate coincidence giving two bonding and two lone pairs [63]. However, as Scemama et al. demonstrated, the maximum probability domains of bonding electron pairs that are naively assumed to be typically placed in the midbond region can only be found with uncorrelated Hartree-Fock wave functions; as soon as correlated wave functions are used, ... *the bonding pairs separate along the bonds, and 'pre-dissociate'* [62].

In addition to rearrangements due to the PEP, energetic aspects must also be considered. The orbital energy of the $2s$ AO in the carbon atom is about 9 eV lower than that of the $2p$ AOs and therefore the $2s$ AO is always filled before any $2p$ AO is occupied; from boron to fluor the $2s$ AO is doubly occupied by one $\alpha$ and one $\beta$ electron. In carbon, the remaining two valence electrons occupy the triply degenerate $2p$ AOs with identical spins, in accordance with Hund's first rule, giving a $^3P_g$ high-spin ground state. In nitrogen, the three remaining electrons occupy the $2p$ AOs with identical spins resulting in a $^4S_u$ high-spin state. Any further electron must occupy an already singly occupied AO; this is only possible if it has a different spin projection, giving a singlet coupled electron pair. This is what happens in the oxygen atom, but also in nitrogen when an electron is excited from the doubly occupied $2s$ AO. In the carbon atom, however, an electron can be excited from the doubly occupied $2s$ AO into the $2p$ subshell without and with a spin flip. In the first case, the resulting multiplicity is still a triplet, but in the second case, all four electrons have an identical spin; this provides a quintet high-spin state, and the electrons prefer a tetrahedral arrangement. It is noteworthy that the energy of the $^5S_u$ state is only 4.2 eV higher than the energy of the $^3P_g$ state [64]; this is roughly half of the difference of the orbital energies. Although excitation energy must be provided, the repulsion energy in the high-spin state is considerably reduced, first, because the Coulomb repulsion of the electrons in the $2s$ AO, which are not Fermi correlated, is reduced, and moreover the Coulomb repulsion of four tetrahedrally arranged electrons is minimal in the spherical shell. Another consequence of the Fermi correlation is a contraction of the orbitals and thus an increase in the attraction of the electron by the nucleus. All these effects are important when molecules come close and the Pauli repulsion between them increases. The increase in the inter-molecular distance reduces it, but if this is not possible, changes from local low-spin to local high-spin arrangements in the interacting molecules can reduce the Pauli repulsion. In any case, energy is needed for the excitation, and, moreover, something must trigger the spin flip. These aspects were already discussed in the studies of the dissociation of ethene, disilene and silaethene [54,55,58]; the insertion reaction of methylene and silylene into the hydrogen molecule [54,55]; and the addition of methylene and silylene, respectively, to ethene, disilene and silaethene, respectively [65].

In the ground state of the dissociated $C_2$ system, both carbon atoms are in their $^3P_g$ ground states. The coupling of the atoms gives 18 molecular terms, 6 singlets, 6 triplets, and 6 quintets. Among the singlets are two $\Sigma_g^+$ states and one $\Delta_g$ state. Only these states are responsible for the stabilization of the system when the atoms approach; the large weight of LC05 suggests that the bonding situation is dominated by a $\sigma$ and a $\pi$ bond, but the weight of LC06, which represents two $\pi$ bonds without a $\sigma$ bond, shows that even at long distances the number and kind of bonds is not definite. At short C–C distances, the weight of LC07 becomes large; this LC describes two singlet coupled atomic quintet states. The weight of LC07 is larger than those of LC05 and LC06, but ionic LCs or LCs describing intra-atomic charge shifts contribute together much more to the ground state wave function than LC07. The attempt to claim that $C_2$ has a quadruple bond around the equilibrium distance ignores the fact that the occurrence of LC07 at the equilibrium does not mean that the carbon atom is there in a local quintet state. After all, no interacting subsystem of a system is in a pure state but only in a mixed state, which allows one only to say with which probability a certain pure state can be expected. One can obtain this information from the reduced density matrix for the subsystem considered. Moreover, this

information will be very different from what many scientists expect; they want to describe the bonding situation using concepts that are not compatible with electron structures that must be described by multi-configurational wave functions.

## 6. Method

All calculations were made with CAS(n,n) wave functions, where $n$ active electrons are distributed among the same number of active orbitals, the wave functions are linear combination of configuration state functions (CSF) generated with the GUGA technique. All calculations were conducted with a local version of GAMESS [66]. For all systems but ethane, the cc-pVTZ basis set was used, and the ethane system was calculated with the cc-pVDZ basis. The single bond in ethane is represented by a CAS(2,2) wave function, the double bond in ethene by a CAS(4,4) wave function, and the triple bond in $N_2$ by a CAS(6,6) wave function. The electron distribution in $C_2$ is described by a CAS(8,8) wave function. In all systems, the two lowest MOs (positive and negative linear combination of $1s$ AOs) are kept frozen. The potential energy curves were calculated at a grid of equidistant points; for ethane and ethene, the distances between nearest neighbours are 0.1 Å for $N_2$ and $C_2$, the distances are 0.05 Å. The geometries of ethene and ethane were optimized for each frozen C–C distance. The fragments of the four systems are the C and the N atoms for $C_2$ and $N_2$, and the methyl radical is for ethane and methylene for ethene. The fragment wave functions were calculated for high spin states using low level methods, e.g., UHF; the methyl and methylene geometries were taken from the optimized molecular geometries. For each bond length, the optimized CASSCF MOs are localized on the respective fragments using an orthogonal Procrustes transformation [53]. Doubly occupied non-active MOs are transformed into doubly occupied fragment MOs (FMO), which are delocalized in case of methyl and methylene; active MOs are transformed into FMOs that resemble AOs or hybrid AOs. The CSFs constructed with these FMOs are dubbed OVB CSFs (orthogonal valence bonds). Finally, the CI matrix is set up with the OVB CSFs and diagonalized. This provides the energies and weights for all OVB CSFs.

**Funding:** Open Access Funding by the University of Graz.

**Data Availability Statement:** Not applicable.

**Acknowledgments:** The author acknowledges helpful comments by W.H.E. Schwarz and by the reviewers.

**Conflicts of Interest:** The author declares no conflict of interest.

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
