# Peer review of "Chemical Bonding in the C2 Molecule"

_inorganics, doi:10.3390/inorganics11060245_

Round 1

Reviewer 1 Report

As the title indicates, this paper is aimed at elucidating the nature of chemical bonding in the C2 molecule. There has been a lot of papers dealing with this topic, but the bonding models presented by these various papers are not at all discussed in the present manuscript, and only one of these papers (Ref. 43) is cited !

Even more troublesome: the present paper leads to no conclusion at all about bonding in C2. There is just one sentence in the abstract saying that the present study does not allow one to decide if C2 has a double, triple or quadruple bond.

What is the interest of publishing a theoretical study that is recognized to lead to no conclusion at all ? This paper does not bring any insight to the question at hand, and therefore should not be published.

Author Response

If the reviewer expects a paper saying that there is a double or a triple or even a quadruple bond in C2, then he or she is right: such an answer is not given because such an answer would be dishonest. But this was also also said by many persons involved in the discussion that started after Shaik et al. claimed that C2 has a quadruple bond. The physical origin of bonding in C2 is complex, the detection of the contributions to the stabilization and their description is not yet finally resolved, no folkloristic explanation can be given.

Reviewer 2 Report

In this paper, the authors developed a method to locate the delocalized MOs obtained from CASSCF on predefined fragments using orthogonal transformation to obtain the orthogonal fragment MOs. The bonding situation in ethane, ethene, and nitrogen molecules have been studied to elucidate bond dissociation. The examples provided serve as a strong validation for the proposed methodology by the author. Currently, the issue lies in the fact that the author's examples solely pertain to homo-atomic bonds. Further verification is required to determine whether this method accurately describes covalent bonds composed of different atoms.

Author Response

In the revised MS all systems are mentioned that were investigated with OVB. C2 is the only homo-nuclear diatomic molecule studied up to now, other systems are chosen to study the differences in bonding between elements of the second and the third period, in detail, carbon and silicon.

The next paper will study the elimination of carbene-like species from three rings with carbon and silicon atoms.

Reviewer 3 Report

See file

Author Response

Ad 1) I modified the text and stressed that in crystals with many-electron ions the fermionic character is mainly responsible for the repulsion of the ions. But the effect of the Pauli exclusion principle can only approximately covered by empirical potentials like Lennard-Jones or Bohr. The only elementary interaction between charges particles is, after all, the Coulomb interaction.

A very good introduction to empirical Pauli repulsion models can be found in Rackers and Ponder,  J. Chem. Phys. 150, 084104 (2019); doi: 10.1063/1.5081060

2) I cannot see how this discussion would improve the understanding of orthogonal AOs. Neither would a discussion of the limit R->0 for the normalized spatial triplet function ab-ba.

3) The FCI of a CAS wave function covers all couplings that result from the distribution of all active electrons among all active MOs. If the MOs are localized to the fragments, all local fragment states that can be constructed with the valence fragment orbitals are automatically created. The weights of coupled atomic states are obtained from the OVB calculation. If the weight is large, an OVB CSF can be discussed, if it small, it is not discussed. As a rule: coupled local excited states have small weights, if the excited states are low-spin states. For example, coupled excited singlet states of methylene or silylene have much smaller weight than the corresponding coupled triplets.

Since the list of OVB CSFs is GUGA generated but not hand-selected, nothing is left out that intentionally.

4) Carbene is replaced by methylene.

5) Done.

Typos and other kind of sloppiness.

1) Done

2) Done

3) Can't see where this should come from. Not from the .bib file.

4) Done

Bib data. I was happy to get .bib data from various sources, sometimes with sometimes without DOI. I did not look for missing DOIs.

Reviewer 4 Report

The present manuscript undertakes a detailed study of chemical bonding in the C2 molecule, plus a small number of closely related systems for comparison, based on quantum-chemical calculations. Most of the text is spent on an analysis of which multi-electron configurations, or linear combinations thereof, are important for an accurate qualitative and quantitative description, and how the relative weights change as a function of the C-C distance. In principle, this could be a worthwhile topic, as the small but intrinsically complex C2 molecule is often regarded as a prototypical system for testing traditional notions of chemical bonding with modern computational methods. On the other hand, this means that a large body of literature, comprising highly converged numerical results and detailed chemical interpretations, is already available and must be taken into account for further scientific advances.

Unfortunately, the manuscript is highly deficient in this respect. Although the Introduction extends over three pages, providing a detailed and very knowledgeable overview over early phenomenological concepts of chemical bonding, the historical account stops in the early 1960s, just before computers began to revolutionize academic research, including chemistry. Indeed, the only references to original research after 1963 cited in the Introduction are those by the author himself and by his one-time co-author Klaus Ruedenberg. As a consequence, the manuscript ignores essentially all of the insight obtained from more than half a century of numerical simulations, and proceeds to examine the C2 molecule as if it were a new and unsolved problem. This leads not only to a lack of context but also to obvious omissions. As just one example, the discussion of "the attempt to claim that C2 has around the equilibrium distance a quadruple bond" clearly refers to the much-cited work of S. Shaik et al. [Nature Chemistry 4, 195 (2012)], but there is no reference either to this original work or to the large number of subsequent studies that have sought to clarify the issue.

Crucially, due to the conscious ignorance of context and the current state of the art, the manuscript fails to convey a proper purpose. Although formally submitted as an original research article, there is no clearly formulated question at the end of the Introduction that is still open to the best of current knowledge, and there are also no final conclusions that go beyond other existing publications and provide additional insights. In between, the analysis of the contributions of different linear combinations of configurations is a typical step in convergence checks for quantum-chemical calculations, and has been done many times before. Although it is true that the author has his very own brand of orthogonal atomic orbitals and cannot rely on published results for other basis sets, there is no convincing argument that this method is in any way superior or provides more insight. As such, the analysis of the importance and weights of various configurations is more suitable for a tutorial on quantum-chemical calculations, while some of the extended discussions and explanations could well feature in an opinion or review, if they included a larger overview.

In short, it is laudable that the author repeats some elementary quantum-chemical calculations for C2 and related molecules in order to draw first-hand conclusions from his own data, but this does not in itself merit a publication unless there is a clear benefit for readers and additional insight that has not been reported before. For this purpose, the author must revise the manuscript to provide a proper account of the current state of the art, identify clear research questions that remain open in the light of previous publications, and draw convincing conclusions that constitute an advance in this respect. As a guiding question, if "there is an ongoing debate on the bond order in C2" as of 2023, despite a wealth of previous studies, what argument will convince readers to perceive this manuscript as a significant step in resolving the debate rather than as another particular result, like many others before? Unless the purpose of the manuscript and its actual place in the context of current quantum-chemical research are clear, publication cannot be recommended.

Notwithstanding the criticism above, the manuscript is very well written and provides an informative and very readable account of historical notions of chemical bonding as well as a detailed description of a well-converged quantum-chemical calculation of a prototypical small molecule.

Author Response

The criticism was absolutely justified. I changed large parts of the Introduction and included the major arguments in the debate on quadruple bonding, I also stressed that in my opinion spin arrangements in the carbon atoms, the rearrangement during the reaction are crucial for the understanding of covalent bonding.  The comparison of VB and OVB is discussed in a separate section, I criticize the use of ``covalent'' for geminals or wave functions in general, because it includes a change in the weights of neutral and ionic contributions, which is not reflected in the form of the Heitler-London wave function.

And I made some minor changes in the presentation of the results and in the Discussion.

Reviewer 5 Report

Useful analysis

Author Response

Thank you.

Round 2

Reviewer 1 Report

The author argues that he has the honesty to say that his results are not conclusive. While in my first report I wrote that inconclusive results do not deserve publications, I now reckon that the honesty of the author is a virtue that is not shared by other authors, like e.g. in refs. 33-36. Since these latter authors got their papers published while using inappropriate methods and still allowing themselves to draw polemic conclusions, I guess the present paper is at least better and therefore can be published as is.

Reviewer 4 Report

The principal criticism from the first round of review reports centred on the fact that the manuscript made too little contact with current works by other authors and referred to ongoing debates only indirectly rather than in the form of clear attributions and citations. As a consequence, the place of this manuscript within these debates and its relation to timely issues of investigation remained obscure, especially as there were also no clear-cut conclusions. On the other hand, the way how the research was carried out and how the results were analyzed was already regarded as appropriate, and the presentation stood out as a positive example that provided much insight for readers.

In the revised version, the author has expanded the Introduction substantially, so that more recent computational works concerning the bondign situation in the C2 molecule are also covered. Considering the very dynamic development of this field, the number of newly added references is not large, but the most essential sources relevant to the question at hand are properly cited. In this way, the Introduction now fulfills its role of outlining the present state of the art, together with the open questions that remain to be solved, in a way that is clearly understandable for readers, and thereby sets the stage for the subsequent investigation. Although it is, of course, unfortunate that the manuscript does not provide an answer to the bond order of C2, it must be accepted that this is not a clear-cut case. At least, the issues and problems are clearly discussed, so that the manuscript still constitutes a benefit for readers and a helpful contribution in this debate. In summary, therefore, I recommend to accept the manuscript in its present, revised form.